# On the Role of Noise in Factorizers for Disentangling Distributed Representations

**Geethan Karunaratne**  **Michael Hersche**  **Abu Sebastian**  **Abbas Rahimi**

IBM Research – Zurich
michael.hersche@ibm.com, {kar,ase,abr}@zurich.ibm.com

## Abstract

One can exploit the compute-in-superposition capabilities of vector-symbolic architectures (VSA) to efficiently factorize high-dimensional distributed representations to the constituent atomic vectors. Such factorizers however suffer from the phenomenon of limit cycles. Applying noise during the iterative decoding is one mechanism to address this issue. In this paper, we explore ways to further relax the noise requirement by applying noise only at the time of VSA's reconstruction codebook initialization. While the need for noise during iterations proves analog in-memory computing systems to be a natural choice as an implementation media, the adequacy of initialization noise allows digital hardware to remain equally indispensable. This broadens the implementation possibilities of factorizers. Our study finds that while the best performance shifts from initialization noise to iterative noise as the number of factors increases from 2 to 4, both extend the operational capacity by at least $50\times$ compared to the baseline factorizer resonator networks. Our code is available at: https://github.com/IBM/in-memory-factorizer

## 1    Introduction

Some basic Visual perception tasks, such as disentangling static elements from moving objects in a dynamic scene and enabling the understanding of object persistence, depend upon the factorization problem [1–4]. The principle of factorization extends to auditory perception, e.g. separating individual voices or instruments from a complex soundscape [5]. Factorization also plays a key role in higher-level cognitive tasks. Understanding analogies for example requires decomposing the underlying relationships between different concepts or situations [6–10]. While biological neural circuits solve the above challenges deftly, factorization remains a problem unsolvable within polynomial time complexity [11]. Outside the biological domain, factorization is at the core of many rapidly developing fields. In robotics, factorization can enable robots to 1) understand their environment by parsing visual scenes and identifying objects, locations, and relationships [12], and 2) plan and execute tasks by decomposing complex actions into a simple sequence of steps. Factorizing semi-primes has implications for cryptography and coding theory [13]. Factorization helps develop more transparent and explainable AI systems [14], by decomposing complex decision processes into understandable components.

Vector-symbolic architectures (VSA) [15–18] is an emerging computing paradigm that can represent and process a combination of attributes using high-dimensional holographic vectors. Unlike the traditional methods where information might be stored in specific locations (e.g., a single bit representing a value), VSA distributes the information across the entire vector signifying a holographic nature. This means if some components of the vector are corrupted or lost, the overall information can still be recovered due to the distributed nature of the encoding. This makes VSA an ideal candidate for realization on low signal-to-noise ratio in-memory computing fabrics based on e.g., resistive

38th Second Workshop on Machine Learning with New Compute Paradigms at NeurIPS 2024 (MLNCP 2024).

RAMs [19, 20] and phase-change memory [21]. In high-dimensional spaces, randomly generated vectors are very likely to be nearly orthogonal aka quasi-orthogonal. This crucial property allows efficient and robust representation of a vast number of different concepts in VSA with minimal interference using these nearly orthogonal vectors as building blocks. More details on the background of VSA are provided in Appendix 5.1, and interested readers can refer to survey [22, 23].

A typical algebraic operation for combining $F$ different attributes in an object is to element-wise multiply associated $D$-dimensional holographic vectors. Due to the properties of high-dimensional spaces, this operation tends to produce unique representations for different attribute combinations. A deep convolutional neural network can be trained to generate these product vectors approximately [14] by taking the raw image of the object. The inverse of the binding problem becomes the factorization problem which is the disentangling of an exact product vector or, as in the latter case, an inexact product vector, into its constituent attribute vectors. While binding is a straightforward calculation, factorization involves searching for the correct combination of attributes among an exponentially large space of possibilities.

Resonator network [24, 25], a type of factorizer, built upon the VSA computing paradigm, introduces a promising approach to perform rapid and robust factorization. Resonator networks employ an iterative, parallel search strategy over high-dimensional vector data structures called *codebooks*, leveraging the properties of high-dimensional spaces to explore the search space efficiently and converge on a query's most likely attribute combinations. However, the original resonator networks face some critical limitations. First, the decoding of multiple bound representations in superposition poses a challenge due to the noise amplifications introduced in traditional explaining-away methods. Therefore, a recent work [26] expanded the pure sequential explaining-away approach by performing multiple decodings in parallel using a dedicated query sampler, which improved the number of decodable bound representations.

As a second limitation, resonator networks' iterative search strategy *limit cycles* poses an obstacle to effective functioning. As one potential remedy to break free of limit cycles during the iterative decoding, the IN-MEMORY FACTORIZER (IMF) [27] harnesses the intrinsic stochastic noise of analog in-memory computing (IMC). Together with an additional nonlinearity in the form of a computationally cheap thresholding function on the similarity estimates, IMF solves significantly larger problem sizes compared to the original resonator networks. Indeed, more recent advancements on resonator networks [28] make also use of nonlinearities in the form of ReLU and exponentiation and noise on the similarity estimates. In this case, the noise has to be generated by a dedicated noise source (e.g., a random number generator) at every decoding iteration. IMF however does not need such an additional noise source thanks to the IMC's intrinsic noise; yet, its performance relies on the availability of underlying hardware with specific noise levels across decoding iterations.

In this article, we explore novel factorizers with alternative noise requirements to mitigate limit cycles. In particular, we propose ASYMMETRIC CODEBOOK FACTORIZER (ACF), where codebooks are initialized with some noisy perturbations and the same instance of noise used across iterations providing a relaxed noise requirement to circumvent limit cycles. We conceptualize a purely digital factorizer design that can provide energy efficiency gains by eliminating data conversions. We find that compared to the baseline resonator network [24], the variants embracing noise, IMF, and ACF, always perform better in terms of operational capacity and require fewer iterations to converge at different sizes of search spaces.

## 2 Background: Resonator Networks

The resonator network is an iterative algorithm designed to factorize high-dimensional vectors [24]. Essential to the operation of resonator networks are *codebooks*, which serve as repositories of quasi-orthogonal high-dimensional vectors called *codevectors*, with each codevector representing a distinct attribute value. For instance, in a system designed for visual object recognition, 3 separate codebooks might store representations for shapes (like circles, squares, triangles), colors (red, green, blue), and positions (top left, bottom right, center) each having 3 distinct possibilities for the attribute values. See Fig. 1. The number of potential combinations of these codevectors to form product vectors grows $M^F$ with respect to the number of attributes (i.e., factors $F$) and attribute values (i.e., codebook size $M$) exponentially. However, the dimensionality ($D$) of these vectors is usually fixed to several hundred leading to ($D << M^F$), searching for the specific combination of codevectors that compose a given product vector becomes computationally expensive, presenting a hard combi-

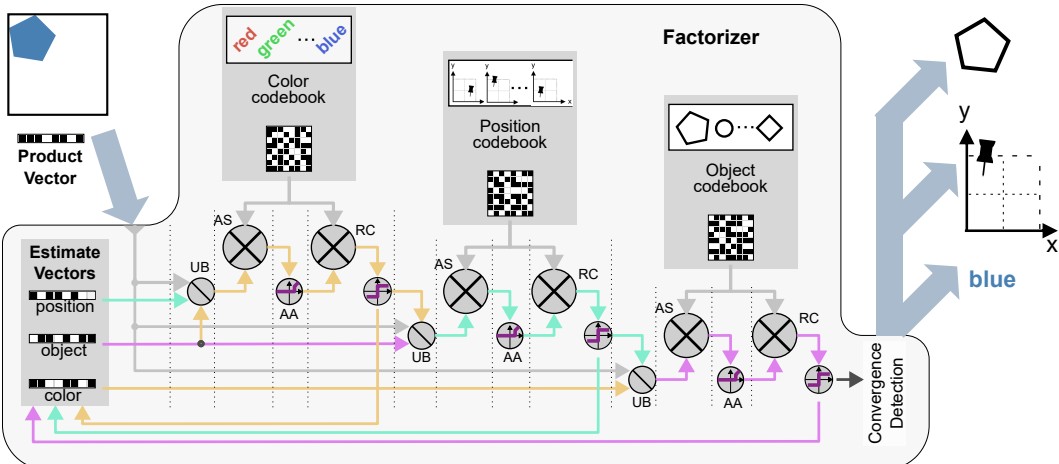

Fig. 1: Factorization of perceptual representations: color position and object type using factorizers. Taking the product vector as input, and starting from initial estimate vectors the factorizer undergoes unbinding (UB), associative search (AS), attention activation (AA), and reconstruction (RC) phases to iteratively refine the estimate. AS and RC take the biggest share of the computing and memory. They map to a predominantly MVM operation.

natorial search problem. Thanks to the quasi-orthogonal property of codevectors however, resonator networks can rapidly navigate the vast search space.

Starting with an initial estimate for each factor, and the product vector, it updates the estimates one factor at a time. It achieves this by first *"unbinding"*(UB) or removing the influence of the estimated values of all but the selected factor from the product vector. In the context of bipolar spaces, where codevector components are +1 or -1, unbinding is accomplished through element-wise multiplication. Secondly, the unbound vector is compared against all codevectors within the corresponding codebook using an associative search (AS), typically a series of cosine similarity or dot products that can be formulated as a matrix-vector-multiplication (MVM) operation. The associative search outputs an attention vector of length $M$, measuring how the unbound vector aligns with the codevectors. Thirdly, the attention vector is passed through an optional attention activation (AA) function. This allows us to filter out uninteresting codevectors. Fourthly, using activated attentions as weights, codevectors are superimposed to reconstruct a new estimate vector. This reconstruction (RC) operation is an MVM operation with the transpose of the codebook itself acting as the reconstruction matrix and the activated attention acting as the vector. The four phases are repeated, refining the estimates for each factor with each iteration until the resonator network converges to a solution representing the most probable factorization of the product vector.

This can be mathematically formulated as follows:

Let us consider a three-factor ($F = 3$) problem with the factors originating from three codebooks:

$\mathbf{A} = \{\, \mathbf{a}^{(1)}, \mathbf{a}^{(2)}, \cdots, \mathbf{a}^{(M)} \,\}, \quad \mathbf{B} = \{\, \mathbf{b}^{(1)}, \mathbf{b}^{(2)}, \cdots, \mathbf{b}^{(M)} \,\}, \quad \mathbf{C} = \{\, \mathbf{c}^{(1)}, \mathbf{c}^{(2)}, \cdots, \mathbf{c}^{(M)} \,\},$ each with $D$ dimensional bipolar codevectors $\mathbf{A}, \mathbf{B}, \mathbf{C} \in \{-1, +1\}^{D \times M}$. Let the estimate for each factor be represented using $\hat{\mathbf{a}}, \hat{\mathbf{b}}, \hat{\mathbf{c}}$ respectively, and the product vector denoted with $\mathbf{x}$, then for the first factor, unbinding of other factors is given by: $\tilde{\mathbf{a}} = \mathbf{x} \oslash \hat{\mathbf{b}} \oslash \hat{\mathbf{c}}$. Based on the unbound vector $\tilde{\mathbf{a}}$ of the first factor, AS, AA, and RC phases can be written as in the following equations respectively.

$$\boldsymbol{\alpha}_a(t) = \tilde{\mathbf{a}}(t)\mathbf{A}^T \quad (1) \qquad , \boldsymbol{\alpha}'_a(t) = f(\boldsymbol{\alpha}_a(t)) \quad (2) \qquad , \hat{\mathbf{a}}(t+1) = sign(\boldsymbol{\alpha}'_a(t)\mathbf{A}) \quad (3)$$

Where $t, \boldsymbol{\alpha}, f, sign$ stands for iteration number, attention vector, activation function, and signum function respectively. A similar approach can be followed to compute the next estimate vector of the other factors $\hat{\mathbf{b}}, \hat{\mathbf{c}}$.

# 3   Breaking Free from Limit Cycles with Noise

As we discussed, resonator networks operate iteratively, progressively refining their estimates for each factor by comparing them to codebook entries. However, during this iterative process, the network can get trapped in a repeating sequence of states. Then the network's estimate for a factor oscillates between a small set of codevectors without ever settling on the true factor. This phenomenon, referred to as a *limit cycle*, can prevent the network from reaching the optimal solution.

The emergence of limit cycles can be attributed to the symmetric and deterministic nature of the codebooks used in the AS and RC phases of the baseline resonator network's (BRN) [24, 25] search procedure. This deterministic behavior is particularly problematic when the search space is large and contains many local minima, which can trap the network's updates in a repeating pattern. When a resonator network gets stuck in a limit cycle, it fails to converge to the correct factorization, even if given an unlimited number of iterations. This lack of convergence can significantly impact the network's accuracy and efficiency, rendering it ineffective for tasks that require precise factorization.

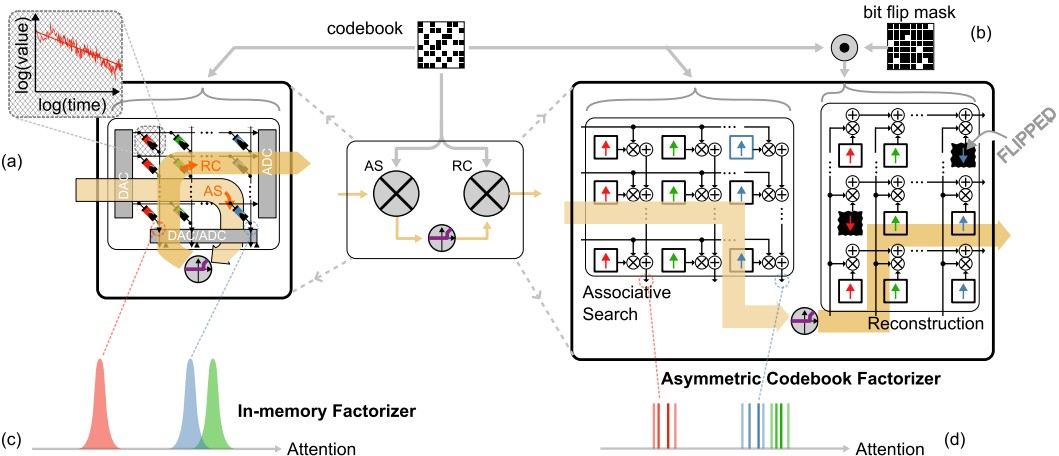

Fig. 2:   Implementing noise during a decoding iteration of a single factor of factorizer. (a) The codebooks are implemented on an analog memory device crossbar array which introduce intrinsic noise to each iteration of both the AS and RC phases. (b) The codebooks are implemented on digital memory devices. The second codebook used in the RC phase is made asymmetric from the first codebook in the AS phase by a bit flip mask perturbation. (c) The resulting attention using analog IMC. It can have a continuous distribution. (d) The resulting attention using asymmetric codebooks. It follows a discrete distribution.

Introducing stochasticity into the network's update rules can help it break free from deterministic limit cycles. This is a crucial finding that not only pushes the factorizers' operational capacity but also shifts the hardware landscape they thrive. In particular, there is a potential for leveraging the intrinsic randomness associated with memristive devices in IMC implementations of factorizers as prescribed in IMF [27]. An example implementation is illustrated in Fig. 2(a). The codevectors of a codebook are programmed along the columns of the crossbar arrays. In the first crossbar used for the AS phase, the inputs are passed through the west side digital to analog converters (DACs), and the resulting currents/charges are collected on the south periphery and converted back to the digital domain using the analog to digital converters (ADCs). After activating the resulting attentions, the sparse attention vector is input through the south-side DACs for the RC phase. The resulting currents/charges are collected through the east side periphery and converted to digital using ADCs before converting via signum function to the next estimate vector.

The memory devices used in these arrays are fabricated using phase-change memory (PCM) technology and exhibit natural variations in their behavior forming a near-Gaussian distribution of the attention result as seen in Fig. 2(c). This can be expressed in an approximated form as $\boldsymbol{\alpha}_a(t) = \tilde{\mathbf{a}}(t)\mathbf{A}^T + n$. Where $n$ is the $M$-dimensional noise vector sampled from i.i.d. Gaussian distribution $n \sim \mathcal{N}(0, \sigma \cdot \mathbf{I}_M)$ and $\sigma$ denotes the standard deviation of the noise, which ranges around 0.01 in a recent large-scale chip based on phase-change memory devices [29]. As seen in

Sec. 4, this $\sigma$ value falls in the *useful noise* range enabling escaping repeating patterns and exploring a wider range of solutions.

Even if an arbitrary number of iterations are allowed, a deterministic digital design with unperturbed symmetric codebooks fails to achieve the accuracy of the IMF due to its susceptibility to limit cycles. Stochasticity, as a key operation to eliminate limit cycles, has significant added costs in terms of energy and area in mature digital hardware implementation. For example, generating Gaussian noise involves several expensive floating point operations such as exponentiation and multiplication.

We explore the possibility of inserting the noise into the codebooks at the time of initialization. If the same noise is added to both copies of the codebook it would still keep the same quasi-orthogonal relationship of the codevectors and would not change the dynamics of the factorizer. Instead, we propose perturbing only a single copy of the codebook using a randomly generated bitflip mask. Fig. 2(b) shows perturbing the codebook used in the RC phase by applying a bit flip mask $BFM \in \{-1, +1\}^{D \times M}$ of certain sparsity as shown in Eq. 4. We call this type of model an Asymmetric Codebook Factorizer (ACF).

$$BFM(r) = \begin{cases} +1 & \text{if } \mathbf{u} + r > 1 \\ -1 & \text{otherwise} \end{cases} \qquad \begin{aligned} \mathbf{A}_{RC} &= \mathbf{A} \odot BFM(r) \\ \hat{\mathbf{a}}(t+1) &= sign(\boldsymbol{\alpha}'_a(t)\mathbf{A}_{RC}) \end{aligned} \qquad (4)$$

Where $\mathbf{u} \sim \mathcal{U}(0,1) \in [0,1]^{D \times M}$ is a noise matrix sampled from the uniform distribution. The sparsity $r$ is a hyperparameter that can be optimally obtained using a hyperparameter search scheme. The perturbed codebook for the RC phase $\mathbf{A}_{RC}$ is calculated by element-wise multiplication between $BFM$ and the codebook as shown in Eq. 4. As $r$ increases the factorizer starts losing its ability to converge and iterate on the correct solution. However, with the presence of the convergence detection circuit detailed in Appendix 5.2, the need to *resonate* is not essential. Similarly, codebooks that are not bipolar in nature [30] can be perturbed using an appropriate random function.

Apart from noise, another known approach to solving limit cycles and converge faster to the right solution involves non-linear activation functions. One such activation function, employed both in IMF and ACF, uses a threshold to sparsify the attention vector. It is explained further in Appendix 5.3.

## 4   Results and Discussion

We conduct software experiments to measure the operational capacity and other behaviors of different variants of factorizers, namely the BRN, IMF, and ACF. Operational capacity is defined as the maximum size of the search space that can be handled with more than 99% accuracy while requiring fewer iterations than what a brute force approach would have taken. A brute force approach would in the worst case find the correct factors in $M^F$ steps.

The results are presented in Fig. 3. We consider 3 cases for the number of factors $F = \{2, 3, 4\}$. The dimensions of the codevectors for these cases are set based on the minimum dimensions reported in the BRN, namely $D = 1000, 1500, 2000$ respectively for $F = 2, 3, 4$ respectively. For each case, we span the search space size starting from $10^4$ up to a maximum of $10^{11}$ until the operational capacity is reached. At each search space size, we conduct 5000 trials factorizing randomly sampled product vectors. We calculate the average accuracy and the average number of iterations.

The BRN reaches its capacity at approximately $10^5$, $10^6$, and $10^7$ for $F = 2, 3, 4$ cases respectively. Although the accuracy momentarily dropped slightly below 99% in a few search space sizes between $10^5$ and $10^6$ at $F = 2$, IMF does not reach the operational capacity for all search space sizes tested. For ACF, we observe the reaching of the operational capacity at $> 5 \times 10^9$ for $F = 4$. In the other two cases, ACF did not reach the operational capacity point for the search space sizes tested. The momentary drop in accuracy and rise in iterations in certain search spaces can be attributed to inadequate hyperparameter exploration. The optimum hyperparameter setting we achieved during our experiments is further detailed in Appendix 5.4

Both IMF and ACF exhibit better performance in terms of operational capacity and the number of iterations compared to the BRN. In theory, the IMF has better control over noise as it is applied over the iterations. This becomes clear in the $F = 4$ case where it outperforms ACF by achieving greater operational capacity. ACF however edges over IMF in the $F = 2$ case where there are fewer interactions among factors.

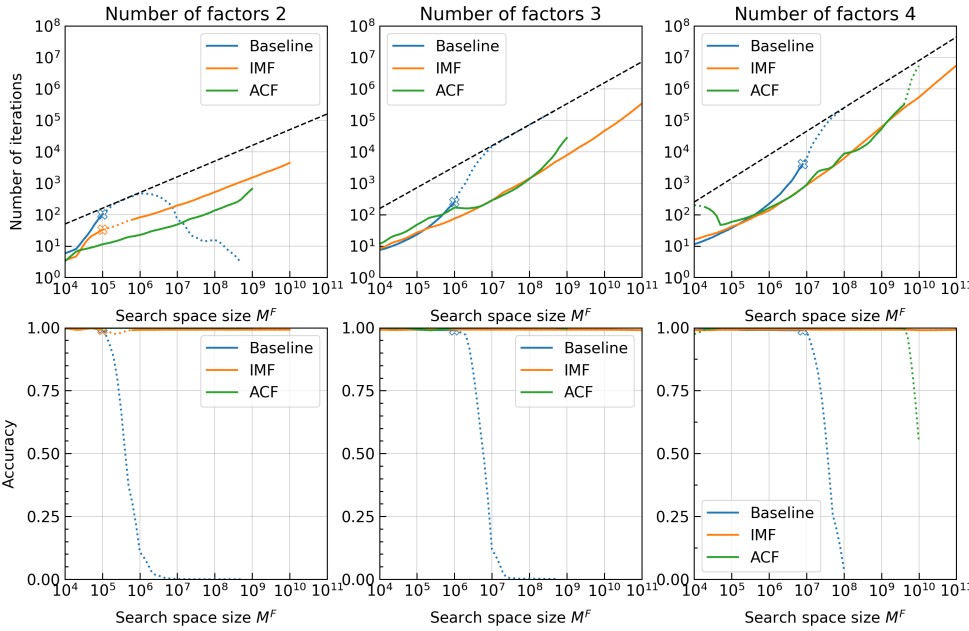

Fig. 3: The number of iterations (top row) and accuracy (bottom row) for three variants of the factorizer: baseline resonator (in blue) [24], in-memory factorizer (in orange) [27], and our asymmetric codebook factorizer (in green). The left, center, and right column results correspond to 2,3, and 4-factor scenarios, respectively. The regions that meet operational capacity criteria (i.e. $\geq 99\%$ accuracy) are plotted with solid lines while other regions are plotted with dotted lines.

Another aspect to consider is the hardware design costs. As shown in Fig. 2, IMF achieves area efficiency with a single copy of codebooks in a crossbar array and achieves energy efficiency by performing arithmetic operations implicitly using device physics. ACF on the other hand has explicit multipliers and adders but saves the bulk of the energy spent on converting data from digital to analog and vice versa several times per decoding iteration. As a consequence of the converter-less design, ACF can operate faster, with several nanoseconds per iteration as opposed to several microseconds. Thus ACF can achieve more iterations per unit period of time.

The principles used in the IMF and ACF are not mutually exclusive. While in this work we study and compare their standalone performance, there is no reason that prevents them from being employed in unison. One possible realization of this involves perturbing the target conductance values corresponding to the codebook values before they get programmed into the analog memory devices in the IMF. Incorporating both sources of notice may result in a synergistic effect enabling higher operational capacity factorizers.

## 5 Conclusion

In conventional wisdom, stochasticity and noise are considered a bane in computing. We demonstrate that factorizers grounded on VSAs empower a new computing paradigm that embraces noise to push the limits of operational capacity. We discuss two variants, the first harnessing intrinsic noise in analog in-memory computing during MVM operation, the second initializing the codebooks with noisy perturbations yielding a model that widely appeals to deterministic digital design systems. While there are tradeoffs, both these variants empirically outperform the baseline resonator networks in multiple facets. When combined with appropriate hardware designs, they provide promising directions to solve factorization problems in large-scale search spaces within reasonable timescales.

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

# Appendix

## 5.1 Vector-symbolic architectures

Here, we provide a brief overview of vector-symbolic architectures (VSAs) [15–18] of which the resonator networks [24, 25] are based on. VSA is a powerful computing framework that is built on an algebra in which all representations are high-dimensional holographic vectors of the same, fixed dimensionality denoted by $D$. This is attributed to modeling the representation of information in the brain as distributed over many neurons. In this work, we consider a VSA model based on bipolar vector space [15], i.e., $\{-1, +1\}^D$. The similarity between two vectors is defined as the cosine similarity:

$$\text{sim}(\boldsymbol{x}_1, \boldsymbol{x}_2) = \frac{\langle \boldsymbol{x}_1, \boldsymbol{x}_2 \rangle}{||\boldsymbol{x}_1|| ||\boldsymbol{x}_2||} = \frac{\langle \boldsymbol{x}_1, \boldsymbol{x}_2 \rangle}{D} \tag{5}$$

As one of the main property of the high-dimensional vector space, any two randomly drawn vectors lie close to quasi-orthogonality to each other, i.e., their expected similarity is close to zero with a high probability [18]. The vectors can represent symbols, and can be manipulated by a rich set of dimensionality-preserving algebraic operations:

- Binding: Denoted by $\odot$, the Hadamard (i.e., element-wise) product of two input vectors implements the binding operation. It is useful to represent a hierarchical structure whereby the resulting vector lies quasi-orthogonal to all the input vectors. The binding operation follows the commutative law $\boldsymbol{x}_1 \odot \boldsymbol{x}_2 = \boldsymbol{x}_2 \odot \boldsymbol{x}_1 = \boldsymbol{p}$.

- Unbinding: The unbinding operation reverses the binding operation. As the element-wise multiplication in the bipolar space is self-inverse, the same operation as for the binding can be used. Using the unbinding operator $\oslash$ the operation is defined as $\boldsymbol{p} \oslash \boldsymbol{x}_1 = \boldsymbol{x}_2$.

- Bundling: The superposition of two vectors is calculated by the bundling operation $\oplus$. The operation is defined by an element-wise sum with consecutive bipolarization. In case of an element-wise sum equal to zero, we randomly bipolarize.

- Clean-up: The clean-up operation maps a noisy vector to its noise-free representation by an associative memory lookup.

- Permutation: Permutation is a unary operation on a vector that yields a quasi-orthogonal vector of its input. This operation rotates the coordinates of the vector. A simple way to implement this is as a cyclic shift by one position.

Interested readers can refer to a detailed survey [22, 23] about VSAs.

## 5.2 Detection of convergence

The iterative factorization problem is said to be converged if, for two consecutive time steps, all the estimates are constant, i.e., $\hat{\boldsymbol{x}}_\text{f}(t+1) = \hat{\boldsymbol{x}}_\text{f}(t)$ for $f \in [1, F]$. We define an early convergence detection algorithm since it avoids unnecessary iterations and in the case of Asymmetric Codebook Factorizer (ACF), the legacy definition of convergence no longer holds.

In the new definition, the factorizer is said to be converged if a single similarity value across all the factors surpasses a convergence detection threshold:

$$\text{converged} = \begin{cases} \text{true}, & \text{if } \max(\boldsymbol{\alpha}_f(t)[i]) > T_\text{convergence} \\ \text{false}, & \text{otherwise,} \end{cases} \tag{6}$$

where $i \in [1, M]$ for $f \in [1, F]$. Upon convergence, the predicted factorization is given by the codevector associated with the largest similarity value per each codebook. This algorithm also eliminates the need to store the history of prior estimates. In the legacy definition of convergence, the previous estimate for each factor had to be stored to be able to compare it to the current estimate and detect convergence, resulting in a total of $F \cdot D$ stored bits. Our experiments show that the optimal convergence detection threshold stays at a fixed ratio of $D$ for any given set of hyperparameters and problem sizes.

## 5.3 Threshold-based Activation

Replacing the identity function, which acts as a linear activation in the BRN, with a nonlinear winner-take-all approach can enhance both the convergence rate and the maximum solvable search space[27]. This strategy sparsifies the similarity vector, essentially zeroing out weaker similarity values and focusing the network's attention on the most promising candidates. By suppressing less likely solutions, sparse activation functions can help prevent the network from getting bogged down in local minima and facilitate convergence to the global optimum. For a threshold $T$, the threshold-based attention activation is given as:

$$\forall i \in (1, M) \quad \alpha'[i] = \begin{cases} \alpha[i], & \text{if } \alpha_i > T \\ 0, & \text{otherwise.} \end{cases} \quad (7)$$

## 5.4 Hyperparameter Search

For the three cases of experiments we conducted, we first set the following common hyperparameters for both ACF and IMF. These include dimension ($D$) and search space size ($M^F$). Then the following hyperparameters have to be tuned to achieve the best results. For ACF, they include the sparsity parameter $r$ and the activation threshold ($T$). For IMF, they include iterative noise standard deviation $\sigma$ and the activation threshold ($T$). Tables 1, 2, and 3 provide the optimum hyperparameter combinations that give rise to the best accuracy and number of iterations results reported in Fig. 3.

Table 1: The hyperparameters that achieve the highest accuracy while minimizing the number of iterations for two factors

| Common parameters | | | ACF-specific | | IMF-specific | |
|---|---|---|---|---|---|---|
| Search space size $M^F$ | Number of factors F | Dimension D | BFM Sparsity r | Activation Threshold T | Iterative Noise $\sigma$ | Activation Threshold T |
| 10000 | 2 | 1000 | 0.005 | 0.01 | 0.008 | 0.001 |
| 21609 | 2 | 1000 | 0.075 | 0 | 0.008 | 0.1 |
| 46225 | 2 | 1000 | 0.1 | 0 | 0.008 | 0.1 |
| 99856 | 2 | 1000 | 0.1 | 0 | 0.008 | 0.1 |
| 215296 | 2 | 1000 | 0.1 | 0 | 0.008 | 0 |
| 463761 | 2 | 1000 | 0.1 | 0 | 0.008 | 0 |
| 1000000 | 2 | 1000 | 0.1 | 0.075 | 0.008 | 0 |
| 2155024 | 2 | 1000 | 0.1 | 0 | 0.008 | 0 |
| 4639716 | 2 | 1000 | 0.1 | 0 | 0.008 | 0 |
| 9998244 | 2 | 1000 | 0.1 | 0 | 0.008 | 0 |
| 21548164 | 2 | 1000 | 0.1 | 0.1 | 0.008 | 0.1 |
| 46416969 | 2 | 1000 | 0.1 | 0.1 | 0.008 | 0.1 |
| 1.00E+08 | 2 | 1000 | 0.05 | 0.1 | 0.008 | 0.1 |
| 2.15E+08 | 2 | 1000 | 0.05 | 0.1 | 0.008 | 0.1 |
| 4.64E+08 | 2 | 1000 | 0.05 | 0.1 | 0.008 | 0.1 |
| 1.00E+09 | 2 | 1000 | 0.01 | 0.1 | 0.008 | 0.1 |

Table 2: The hyperparameters that achieve the highest accuracy while minimizing the number of iterations for three factors

| Common parameters | | | ACF-specific | | IMF-specific | |
|---|---|---|---|---|---|---|
| Search space size $M^F$ | Number of factors F | Dimension D | BFM Sparsity r | Activation Threshold T | Iterative Noise $\sigma$ | Activation Threshold T |
| 10648 | 3 | 1500 | 0.1 | 0.01 | 0.007 | 0.001 |
| 21952 | 3 | 1500 | 0.1 | 0.01 | 0.007 | 0.01 |
| 46656 | 3 | 1500 | 0.05 | 0.01 | 0.007 | 0.01 |
| 97336 | 3 | 1500 | 0.05 | 0.01 | 0.007 | 0.01 |
| 216000 | 3 | 1500 | 0.005 | 0.01 | 0.007 | 0.05 |
| 456533 | 3 | 1500 | 0.1 | 0.05 | 0.007 | 0.05 |
| 1000000 | 3 | 1500 | 0.1 | 0.05 | 0.007 | 0.05 |
| 2146689 | 3 | 1500 | 0.1 | 0.05 | 0.007 | 0.05 |
| 4657463 | 3 | 1500 | 0.05 | 0.05 | 0.007 | 0.05 |
| 9938375 | 3 | 1500 | 0.05 | 0.05 | 0.007 | 0.05 |
| 21484952 | 3 | 1500 | 0.01 | 0.05 | 0.007 | 0.05 |
| 46268279 | 3 | 1500 | 0.01 | 0.05 | 0.007 | 0.05 |
| 99897344 | 3 | 1500 | 0.0005 | 0.05 | 0.007 | 0.05 |
| 2.15E+08 | 3 | 1500 | 0 | 0.05 | 0.007 | 0.05 |
| 4.64E+08 | 3 | 1500 | 0 | 0.05 | 0.007 | 0.05 |
| 1.00E+09 | 3 | 1500 | 0 | 0.05 | 0.007 | 0.05 |

Table 3: The hyperparameters that achieve the highest accuracy while minimizing the number of iterations for four factors

| Common parameters | | | ACF-specific | | IMF-specific | |
|---|---|---|---|---|---|---|
| Search space size $M^F$ | Number of factors F | Dimension D | BFM Sparsity r | Activation Threshold T | Iterative Noise $\sigma$ | Activation Threshold T |
| 10000 | 4 | 2000 | 0.1 | 0 | 0.006 | 0.01 |
| 20736 | 4 | 2000 | 0.09 | 0 | 0.006 | 0.01 |
| 50625 | 4 | 2000 | 0.05 | 0 | 0.006 | 0.001 |
| 104976 | 4 | 2000 | 0.02 | 0.001 | 0.006 | 0.001 |
| 234256 | 4 | 2000 | 0.006 | 0.01 | 0.006 | 0.01 |
| 456976 | 4 | 2000 | 0.005 | 0 | 0.006 | 0.01 |
| 1048576 | 4 | 2000 | 0.001 | 0 | 0.006 | 0.01 |
| 2085136 | 4 | 2000 | 0.008 | 0.025 | 0.006 | 0.01 |
| 4477456 | 4 | 2000 | 0.006 | 0.025 | 0.006 | 0.01 |
| 9834496 | 4 | 2000 | 0.006 | 0.025 | 0.006 | 0.05 |
| 21381376 | 4 | 2000 | 0.006 | 0.025 | 0.006 | 0.05 |
| 47458321 | 4 | 2000 | 0.003 | 0.03 | 0.006 | 0.05 |
| 1.00E+08 | 4 | 2000 | 0.002 | 0.03 | 0.006 | 0.05 |
| 2.14E+08 | 4 | 2000 | 0.02 | 0.04 | 0.006 | 0.05 |
| 4.67E+08 | 4 | 2000 | 0.008 | 0.04 | 0.006 | 0.05 |
| 1.00E+09 | 4 | 2000 | 0.008 | 0.04 | 0.006 | 0.05 |

