# OpenReview forum: "On the role of noise in factorizers for disentangling distributed representations"
_NeurIPS.cc/2024/Workshop/MLNCP — MLNCP Poster_

### Official Review · Reviewer_hb9v · 2024-10-03
**Recommendation for acceptance**

**Rating:** 8
**Confidence:** 3

**Review:**

The authors propose a novel approach for factorization in vector-symbolic architecture (VSA) on digital memory devices, mainly aiming at mitigating the limit cycle problem in VSA, which traps the procedure to an undesired local minima. Specifically, they devised a noise injection procedure for a digital memory device involving asymmetric random bit flip, an alternative source of stochasticity to the intrinsic noise utilized to solve factorization by analog memory devices in an existing algorithm. They demonstrated by numerical experiments that the proposed algorithm can perform better when the number of factors is small compared to the existing noise induction method. VSA is an active area of research because it is well suited for non-von-Neumann computing paradigms such as neuromorphic hardware and is expected to be one of the platforms for AI tasks in such platforms. Factorization and its performance are essential parts of VSA because they ensure we can effectively obtain the desired outcome from highly parallel computation that is natural to VSA. As such, I recommend acceptance of this work in MLNCP. That being said, I have a small number of questions and comments that, if addressed, would improve the work's readability.

1) In section 4, the authors show the results of their experiments on BRN, IMF, and ACF. It would help the paper's readability if there was an explanation about whether the experiment was done as a software simulation or on physical hardware.

2) In lines 197-203, the authors discuss the advantage of using digital hardware to overcome digital-to-analog conversion overhead. Because Fig. 3 compares three algorithms based on iteration counts and accuracy, this may hinder readers from understanding the potential of AFC against IMF. It would help understand the advantage of ACF if the expected real-time execution time comparison against IMF is depicted somewhere.

3) In lines 204-209, the authors suggest using IMF and ACF together to make it more performant. It will help readability if the authors explain in what measure the combination becomes "more performant," such as operation capacity, number of iterations, or accuracy.

---

### Official Review · Reviewer_svco · 2024-10-04
**Breaking limit cycles with noise**

**Rating:** 6
**Confidence:** 2

**Review:**

The authors found that introducing stochasticity using noise into the network's update rules can help breaking free from deterministic limit cycles. It may be worth checking whether chaotic dynamics would also contribute to this effect [a].
[a] T. L. Carroll and L. M. Pecora, Using chaos to keep period-multiplied systems in phase, Physical Review E 48, 2426 (1993).

---

### Decision · Program_Chairs · 2024-10-10

Accept (Poster)